# Tighter Lower Bounds on Quantum Annealing Times

Luis Pedro García-Pintos,[1] Mrunmay Sahasrabudhe,[2] and Christian Arenz[2]

[1] *Theoretical Division (T-4), Los Alamos National Laboratory, New Mexico, 87545, USA*
[2] *School of Electrical, Computer, and Energy Engineering,*
*Arizona State University, Tempe, Arizona 85287, USA*

We derive lower bounds on the time needed for a quantum annealer to prepare the ground state of a target Hamiltonian. These bounds do not depend on the annealing schedule and can take the local structure of the Hamiltonian into account. Consequently, the bounds are computable without knowledge of the annealer's dynamics and, in certain cases, scale with the size of the system. We discuss spin systems where the bounds are polynomially tighter than existing bounds, qualitatively capturing the scaling of the exact annealing times as a function of the number of spins.

## I. INTRODUCTION

Quantum annealing is a powerful framework for solving optimization problems on quantum computers [1]. In quantum annealing, one aims to steer the state of a quantum system by properly tailored control functions, from a simple-to-prepare initial state to a target state that encodes the solution to the problem of interest. Typically, the initial and final states are the ground states of an initial and final Hamiltonian $H_i$ and $H_f$. The goal is to design annealing schedules that dynamically transition between both Hamiltonians such that, at some final time $T$, the system's state is close to the desired target ground state. A prominent example of quantum annealing is adiabatic quantum computation. There, the annealing schedule consists of slowly changing the control functions to evolve into the ground state of $H_f$ according to the adiabatic theorem [2]. The scaling of the annealing time $T$ with respect to the system size depends on the complexity of the computational problem to be solved.

Unfortunately, general results that characterize the behavior of $T$ are scarce. The most studied of such results rely on the adiabatic theorem to derive sufficient conditions for successful annealing. The adiabatic theorem shows that the system's final state will be close to the desired ground state of $H_f$ if the protocol is slow enough. Typically, $T$ must scale as $1/\Delta^2$ for adiabaticity to hold, where $\Delta$ is the minimum spectral gap given by the energy difference between the ground and first excited energy eigenstate of the time-dependent Hamiltonian describing the annealer [3]. The spectral gap is typically exponentially small in system size, which leads to exponentially long computing times by adiabatic protocols, noting that necessary conditions for adiabaticity were also studied in Refs. [4] and [5].

However, adiabaticity can be an overly restrictive condition. Often, optimized schedules can be much faster than those guided by the adiabatic theorem (i.e., much faster than $T \sim 1/\Delta^2$). This motivates studying the smallest times $T$ needed in quantum annealing protocols. General lower bounds on quantum annealing times, which hold for non-adiabatic annealing schedules, were derived in Ref. [6]. Leveraging the techniques used in [6], Ref. [7] focuses on the minimum number of rounds needed

to reach a target ground state via an analog optimization algorithm, which is inspired by quantum annealing. The results in Refs. [4–7] give bounds on quantum annealing times given an algorithm's physical implementation (see also Ref. [8], which considers phenomenological limits to quantum annealing times).

The bounds in Refs. [5–7] were derived using techniques usually known as quantum speed limits [9]. A speed limit is a bound on the rate at which a physical quantity can change. Mandelstam and Tamm proved the first speed limit in Ref. [10]. They showed that an observable's expectation value $\langle A \rangle$ satisfies $|d\langle A \rangle/dt|^2 \leq 4\text{Var}(A)\text{Var}(H)$ for any isolated quantum system, where $\text{Var}(A) := \langle A^2 \rangle - \langle A \rangle^2$ and $\text{Var}(H)$ are the variances of an observable $A$ and of the system's Hamiltonian $H$, respectively. Mandelstam and Tamm's result sets a trade-off relation between speed and uncertainty. Reference [5] shows that a similar trade-off relation constrains adiabatic quantum annealing. Meanwhile, Ref. [6] shows a trade-off relation between runtime and quantum coherence in arbitrary (possibly diabatic) annealing.

One shortcoming of recent results that aim to describe the minimum runtime $T$ is that they do not accurately reflect the annealing times in systems where locality matters. For instance, the bounds in Refs. [5–7] are loose for quantum annealing or the quantum approximate optimization algorithm (QAOA) that aims to prepare the ground state of an Ising Hamiltonian [11]; the bounds are independent of the system size, but the actual times typically grow with the system size.

A second shortcoming stems from an inherent property of most speed limits: evaluating the bounds often requires having information on the dynamics, e.g., the path taken in Hilbert space. For instance, evaluating Mandelstam and Tamm's bound $|d\langle A \rangle/dt|^2 \leq 4\text{Var}(A)\text{Var}(H)$ requires knowing $\text{Var}(A)$, which changes as the system evolves. This problem is prevalent in most quantum [12–16] and classical [17–19] speed limits considered in the literature. As such, most quantum speed limits are of limited practical use for accurately capturing the behavior of $T$ in explicitly time-dependent systems, which generally occurs in quantum control and quantum annealing.

In this work, we take steps to address the shortcomings described in the previous two paragraphs by leverag-

ing techniques introduced in [20, 21]. Namely, we derive lower bounds on the annealing time $T$, which in turn implies lower bounds on the circuit depth needed in QAOA. The bounds (i) can be evaluated without information of annealing schedules or state dynamics, and (ii) are tighter than previously considered bounds [5–7] for certain spin models, where the new bounds can scale with the system size.

The manuscript is structured as follows. We begin in Sec. II by deriving simple lower bounds on annealing times that do not depend on the annealing schedule or the system's trajectory through Hilbert space. In Sec. III, we leverage the results from Sec. II to derive speed limits that are tighter than previously considered bounds. We illustrate the scaling with the system size of the new bounds on the analog version of Grover's search algorithm and on a perturbed $p$-spin model. For each example, we compare to existing bounds in the literature. We show that for optimization problems characterized by a well-connected graph with a low-connected vertex, the new bounds display a polynomial improvement over previous results.

## II. FROM SCHEDULE-DEPENDENT TO SCHEDULE-INDEPENDENT SPEED LIMITS

We start by deriving a speed limit based on bounding the Bures distance

$$D_B\left(|\psi(t)\rangle, |\tilde{\psi}(t)\rangle\right) := \sqrt{2\left(1 - \left|\langle\psi(t)| \tilde{\psi}(t)\rangle\right|\right)} \quad (1)$$

between two quantum states $|\psi(t)\rangle = U(t)|\psi_0\rangle$ and $|\tilde{\psi}(t)\rangle = \tilde{U}(t)|\psi_0\rangle$. The unitary transformations $U(t)$ and $\tilde{U}(t)$ are generated by the Hamiltonians $H(t)$ and $\tilde{H}(t)$, respectively and $|\psi_0\rangle$ is the initial state of the system. Along the lines of Ref. [20], in Appendix A we show that Bures distance can be upper bounded by

$$D_B\left(|\psi(t)\rangle, |\tilde{\psi}(t)\rangle\right) \leq \int_0^t dt' \left\|(H(t') - \tilde{H}(t'))\tilde{U}(t')|\psi_0\rangle\right\|, \quad (2)$$

where $\|\cdot\|$ denotes the Euclidean vector norm. Thus, if we choose $\tilde{H}(t) = \langle\psi_0| H(t)|\psi_0\rangle \mathbb{1}$ so that $\tilde{U}(t)$ becomes a global phase, we obtain a lower bound

$$T \geq \frac{D_B(|\psi_T\rangle, |\psi_0\rangle)}{\frac{1}{T}\int_0^T \sqrt{\text{Var}_{|\psi_0\rangle}(H(t))}\, dt}, \quad (3)$$

on the time $T$ to prepare a target state $|\psi_T\rangle \equiv |\psi(T)\rangle$. The bound depends on the variance $\text{Var}_{|\psi_0\rangle}(H(t))$ of $H(t)$ with respect to the initial state $|\psi_0\rangle$. The speed limit in Eq. (3) resembles the one derived by Mandelstam-Tamm in Ref. [10]. As alluded to in the introduction, such bounds are of limited use for time-dependent Hamiltonians since they depend on the control or annealing schedule that prepares the desired target state.

### A. Schedule-independent speed limits on quantum annealing

To derive schedule-independent speed limits that apply to quantum annealing problems, we leverage a technique introduced in [20]. Consider a time-dependent Hamiltonian of the form

$$H(t) = f(t)H_i + g(t)H_f. \quad (4)$$

The goal in quantum annealing is to design control schedules described by the functions $f(t)$ and $g(t)$ that prepare the ground state $|\psi_T\rangle = |E^f_{\min}\rangle$ of the Hamiltonian $H_f$ at some time $T$. The system starts in an initial state $|\psi_0\rangle$, which is typically assumed to be an easy-to-prepare ground state of the Hamiltonian $H_i$. Leveraging the bound (2), we find that the time $T$ to prepare the ground state is lower bounded by

$$T \geq \max\left\{\frac{D_B(|\psi_T\rangle, |\psi_0\rangle)}{g_{\max}\sqrt{\text{Var}_{|\psi_0\rangle}(H_f)}}, \frac{D_B(|\psi_T\rangle, |\psi_0\rangle)}{f_{\max}\sqrt{\text{Var}_{|\psi_T\rangle}(H_i)}}\right\}. \quad (5)$$

Here, $g_{\max} = \max_{t\in[0,T]} |g(t)|$ and $f_{\max} = \max_{t\in[0,T]} |f(t)|$ are the largest amplitudes of the control functions $g(t)$ and $f(t)$, respectively. The first expression is proved by picking $\tilde{H}(t) = f(t)H_i + g(t)\langle\psi_0| H_f |\psi_0\rangle \mathbb{1}$ in (2), in which case the unitary $\tilde{U}(t)$ applied to an initial eigenstate of $H_i$ gives a phase that does not affect Bures distance $D_B(|\psi_T\rangle, |\psi_0\rangle)$. The second bound can be proven analogously by considering the reverse-annealing process.

As suggested by the bound (5), if the control functions are unconstrained, preparing the gound state of $H_f$ can be achieved instantaneously [22], i.e., the ground state can be prepared to arbitrary precision in an arbitrarily short time. Physically, however, energy constraints limit how large $g(t)$ and $f(t)$ can be. Interestingly, even if $f(t)$ is unconstrained but $g(t)$ is limited in amplitude by $g_{\max}$, the ground state cannot be prepared faster than the lower bound (5) implies, and vice-versa. We further note that, unlike Eq. (3), the lower bounds in Eq. (5) are independent of the driving protocol [i.e., independent of the control functions $f(t)$ and $g(t)$].

The lower bound on $T$ scales with the variances of $H_f$ or $H_i$ with respect to the initial or final states. Thus, if one of the variances decreases with system size, the algorithm's runtime must increase accordingly. For instance, if we assume that the initial state is an equal superposition $|\psi_0\rangle = \frac{1}{\sqrt{d}}\sum_{j=1}^d |E^f_j\rangle$ of all eigenstates $|E^f_j\rangle$ of the Hamiltonian $H_f$, then the bound (5) implies

$$T \geq \frac{\sqrt{2d(1 - \frac{1}{\sqrt{d}})}}{g_{\max}\sqrt{\text{Tr}\{H_f^2\}}} \quad (6)$$

where $d$ is the dimension of the Hilbert space. Thus, the time $T$ to prepare the ground state must at least scale

as $\mathcal{O}(\sqrt{d})$ when $\text{Tr}\{H_f^2\} = \mathcal{O}(1)$ remains constant when the system is scaled. This is, for example, the case when $\text{Tr}\{H_f^2\} = 1$ is used as a normalization condition [23, 24]. The $\mathcal{O}(\sqrt{d})$ scaling obtained here is similar to the one found in Ref. [25], where system size-dependent speed limits were obtained for quantum control by estimating the diameter of the quotient spaces that exclude operations that can be achieved instantaneously through control. This scaling was also obtained in Refs. [5–7] for the analog version of Grover's search algorithm, which we consider next to probe the new bound (5).

### B.   Case study: analog Grover search

The aim of Grover' search algorithm is to prepare the state $|m\rangle$ where $m \in \{0,1\}^n$ is a bitstring of size $n$ that corresponds to the solution of an (unstructured) search problem. The $n$-qubit system is initialized in the equal-superposition state $|\psi_0\rangle = \frac{1}{\sqrt{d}}\sum_{x\in\{0,1\}^n}|x\rangle$, where $d = 2^n$. The initial and target states are minimum energy eigenstates of the Hamiltonians $H_i = \mathbb{1} - |\psi_0\rangle\langle\psi_0|$ and $H_f = \mathbb{1} - |m\rangle\langle m|$, respectively. Roland and Cerf proved that an optimized annealing schedule gives a quadratic improvement in runtime [26], i.e., $T = \mathcal{O}(\sqrt{d})$, matching the quantum improvement of Grover's search algorithm over classical algorithms [27]. The optimized annealing schedule derived by Roland and Cerf saturates the lower bound derived in Ref. [6], which shows that going beyond the adiabatic schedule cannot improve the $\sqrt{d}$ scaling.

Assuming ideal annealing where the target state is given by $|\psi_T\rangle = |m\rangle$, and using that $\langle m|\psi_0\rangle = \frac{1}{\sqrt{d}}$, Eq. (5) gives the lower bound

$$T \geq \max\left\{\frac{1}{g_{\max}}, \frac{1}{f_{\max}}\right\}\frac{\sqrt{2(1-\frac{1}{\sqrt{d}})}}{\sqrt{\frac{1}{d}-\frac{1}{d^2}}} = \mathcal{O}(\sqrt{d}), \quad (7)$$

on the time needed to prepare the target state $|m\rangle$ that yields the solution to the search problem. This proves that even if one of the control fields $f(t)$ or $g(t)$ could be made arbitrarily strong, the $\mathcal{O}(\sqrt{d})$ scaling cannot be beaten. The minimum computation time is bounded by the weakest of the control fields. In the case where $M$ bitstrings are solutions to the search problem, i.e., $H_i = -\sum_{j=1}^{M}|j\rangle\langle j|$, $j \in \{0,1\}^n$, for $d \gg M$ the bound becomes $T \gtrsim \max\left\{\frac{1}{g_{\max}}, \frac{1}{f_{\max}}\right\}\sqrt{\frac{d}{M}}$.

In Fig. 1 we compare the exact annealing times with bound (7). The exact annealing times were obtained by numerically optimizing the fidelity error for preparing the ground state using the Broyden-Fletcher-Goldfarb-Shanno (BFGS) algorithm. In the numerical simulations, we assumed that $f(t)$ is unconstrained and used 100 piecewise constant control amplitudes over which we optimize, reporting the average over 100 randomly chosen initial control field values. The results shown in Fig. 1

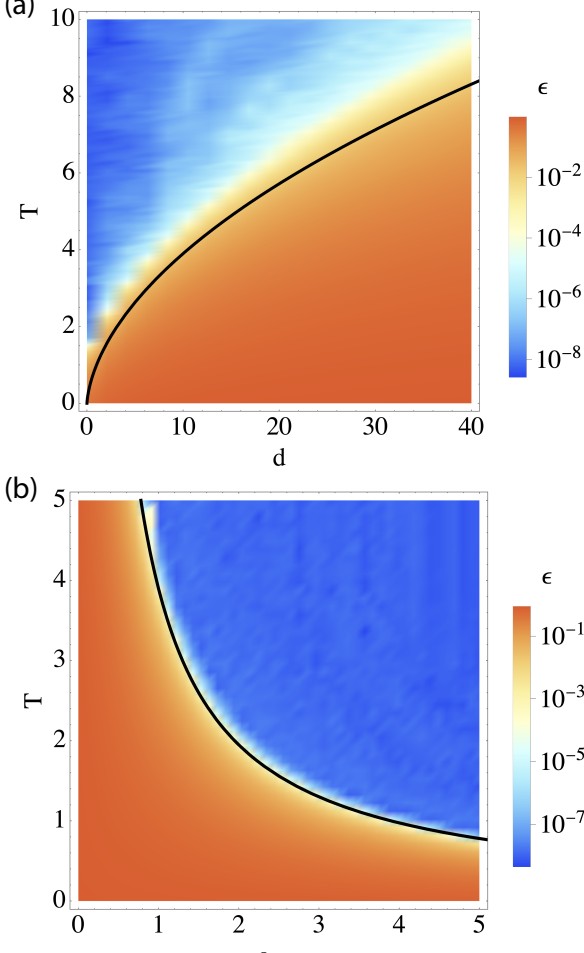

FIG. 1.   Numerical optimization using BFGS to determine the optimal annealing schedule and corresponding annealing times $T$ for the analog version of Grover search. The simulations were performed using 100 piecewise constant control field values averaged over 100 randomly chosen initial control fields. The colormap in (a) shows the fidelity error $\epsilon$ as a function of the system size $d$ and $T$ for a fixed maximum control amplitude $g_{\max} = 1$ while the control field $f(t)$ was assumed to be unconstrained. The colormap in (b) shows $\epsilon$ as function of $g_{\max}$ and $T$ for fixed $d = 10$. In both plots the bound given in (7) is shown as a black curve.

illustrate that the bound (7) shown as a solid black curve is remarkably tight. While asymptotically tight bounds on search by annealing where also derived in Refs. [5, 6], Eq. (7) is slightly more general as it does not restrict the magnitude of the schedule functions.

While the bound (5) captures the annealing time in search problems, it does not accurately reflect the times needed to perform annealing on systems with geometric locality (e.g., spin chains). This is because the Hamiltonian's variances typically grow with system size [28]. This shortcoming is shared with lower bounds on annealing times previously considered in the literature [5–7]. Next, we partially address this shortcoming by deriving

bounds on the annealing time that, in certain instances, grow with system size.

## III. TIGHTER BOUNDS ON ANNEALING TIMES FOR SPIN MODELS

The schedule- and path-independent bound (5) is a special case of a broader class of results that aim to bound quantum control times (i.e., the time to prepare a desired target state or unitary transformation through control fields) by considering choices of $\tilde{H}(t)$ that make the system the most uncontrollable [29]. In fact, the derivation of teh bound (5) rests on comparing the exact evolution $U(t)$ with an evolution $\tilde{U}(t)$ where the control term $g(t)H_f$ is removed. Motivated by the results in [21], we now consider other choices of $\tilde{H}(t)$ to derive tighter bounds on annealing times.

We consider a rotation of $H(t)$ by a unitary transformation $W$ [21], such that $[W, H_i] = 0$. That is, $\tilde{H}(t) = W^\dagger H(t) W = g(t) W^\dagger H_f W + f(t) H_i$, and $W |\psi_0\rangle = e^{i\phi} |\psi_0\rangle$. By relying on the bound (2), we prove in Appendix B that

$$T \geq \frac{D_B(|\psi_T\rangle, W|\psi_T\rangle)}{g_{\max}\|H_f - W H_f W^\dagger\|} = \frac{\sqrt{2}\sqrt{1 - |\langle\psi_T| W |\psi_T\rangle|}}{g_{\max}\|[H_f, W]\|}, \tag{8}$$

where here $\|A\|$ denotes the spectral norm of an operator $A$, and we used that $\|H_f - W H_f W^\dagger\| = \|[H_f, W]\|$ since the spectral norm is unitarily invariant. An analogous result holds by choosing $\tilde{H}(t) = V^\dagger H(t) V$, with $[V, H_f] = 0$ and $V|\psi_T\rangle = e^{i\varphi}|\psi_T\rangle$. We find, $T \geq \frac{D_B(|\psi_0\rangle, V|\psi_0\rangle)}{f_{\max}\|H_i - V^\dagger H_i V\|}$ where $f_{\max} = \max_{t\in[0,T]}|f(t)|$. Combining both bounds yields

$$T \geq \sqrt{2}\max\left\{\frac{\sqrt{1 - |\langle\psi_0| V |\psi_0\rangle|}}{f_{\max}\|[H_i, V]\|}, \frac{\sqrt{1 - |\langle\psi_T| W |\psi_T\rangle|}}{g_{\max}\|[H_f, W]\|}\right\}. \tag{9}$$

In Appendix C, we leverage these results to bound the circuit depth in QAOA.

The bounds in (9) do not depend on the state's path or the annealing schedule. Evaluating them only requires knowledge of (i) the Hamiltonians $H_i$ and $H_f$ that drive the quantum annealing algorithm, (ii) the initial or final states $|\psi_0\rangle$ or $|\psi_T\rangle$, and (iii) the free-to-choose unitaries $V$ or $W$. Next, we show that properly chosen $W$ or $V$ yields speed limits for spin networks that are tighter than previously known bounds [6].

### A. Bounds on annealing times for spin networks

We consider an $n$-spin 1/2 network on a graph $G(V, E)$ with edges $E$ and vertices $V$. We denote by $|V| = n$ and

$|E|$ the number of vertices and edges in the graph, respectively. For concreteness, we consider 2-local systems, where the spin network is described by the Hamiltonian,

$$H_f = \mathcal{N} \sum_{\alpha,\beta\in\{x,y,z\}} \sum_{(i,j)\in E} h_{\alpha,\beta}^{(i,j)} \sigma_\alpha^{(i)} \sigma_\beta^{(j)}. \tag{10}$$

Here, $\sigma_\alpha^{(i)}$ denote Pauli operators $\alpha \in \{x, y, z\}$ acting on the $i$th spin, $h_{\alpha,\beta}^{(i,j)}$ are coupling constants between spins $(i)$ and $(j)$, and $\mathcal{N}$ is a normalization constant. We denote the largest coupling constant in the system by $h_{\max} = \max\{|h_{\alpha,\beta}^{(i,j)}|\}$. The solutions to optimization problems described by the graph $G(V, E)$ can be encoded in the ground states of Hamiltonians like (10) [30].

For the initial Hamiltonian, we take

$$H_i = -\sum_{j\in V} \sigma_x^{(j)}, \tag{11}$$

so that the initial state is $|\psi_0\rangle = |+\rangle^{\otimes n}$, the ground state of $H_i$, where $|+\rangle$ is an eigenstate of $\sigma_x$. We aim to leverage the bound in Eq. (8) to lower bound the time $T$ needed to prepare the ground state of $H_f$ via the time-dependent Hamiltonian $H(t) = f(t)H_i + g(t)H_f$. For simplicity we assume $f_{\max} = g_{\max} = 1$.

In general, the time to prepare the ground state of $H_f$ depends on the magnitudes of the Hamiltonians used to drive the dynamics. For instance, rescaling a Hamiltonian to $H' = aH$ with $a > 1$ trivially decreases time to $T' = T/a$. As the bounds (8) and (9) hold regardless of the rescaling of the Hamiltonian, it is convenient to fix a magnitude of the $H$s when comparing the scaling of $T$ with the system size. For spin networks, we adopt the convention that all Hamiltonians are extensive [31]. That is, $\|H_i\| = n$ and $\|H_f\| = \mathcal{O}(n)$. For geometrically local Hamiltonians, this is ensured by picking the normalization constant $\mathcal{N}$ in $H_f$ to be $\mathcal{N} = \frac{|V|}{|E|} = \frac{n}{|E|}$. This choice ensures that $\|H(t)\| \sim \|H_i\| \sim \|H_f\| = \mathcal{O}(n)$, and that doubling the number of spins in the network approximately doubles the system's energy.

Taking $W = \sigma_\alpha^{(i)}$ in Eq. (8), where $i$ identifies the vertex with the smallest degree $\delta$, it holds that $\|[H_f, W]\| \leq 6\,h_{\max}\,\delta\mathcal{N} = 6\,h_{\max}\,\delta\frac{|V|}{|E|}$, which yields

$$T \geq \frac{\sqrt{2}\sqrt{1 - |\langle\psi_T| W |\psi_T\rangle|}}{6\,h_{\max}\,\delta}\frac{|E|}{|V|}. \tag{12}$$

To illustrate the bound's scaling, consider an almost-complete graph where $|E| = \mathcal{O}(n^2)$ and $|V| = n$, where one node is minimally connected, e.g., $\delta = 1$. Then, the time to prepare the ground state must at least increase linearly with the number of spins $n$ [32]. That is, the lower bound on $T$ grows with $n$ for sufficiently connected graphs where $|E|/|V|$ grows, as long as the smallest vertex degree $\delta$ is small and $|\langle\psi_T| W |\psi_T\rangle|$ is sufficiently small (as we show below, one can often choose $W$

such that $|\langle\psi_T|W|\psi_T\rangle| = 0$). The bound (12) may thus be ideally suited to study long-range systems [33, 34], such as the SYK model [35, 36], where the corresponding graphs have high connectivity.

For instance, consider the minimum time to find the ground state of an Ising Hamiltonian $H_f = |V|/|E|\sum_{(i,j)\in E} h_{z,z}^{(i,j)}\sigma_z^{(i)}\sigma_z^{(j)}$ where the smallest vertex degree is $\delta = 1$. The ground state is given by some bit string $|\psi_T\rangle = |m\rangle$ with $m \in \{0,1\}^n$, so $\langle\psi_T|W|\psi_T\rangle = 0$ for $W = \sigma_x^{(i)}$. Thus, Eq. (12) becomes $T \geq \frac{\sqrt{2}}{h_{\max}}\frac{|E|}{|V|}$, which grows with the graph's connectivity.

We can use the previous example to show that the bound in Eq. (9) is typically tighter than the only schedule-independent bound derived in Ref. [6]. The bound in Ref. [6] is

$$T \geq \tau_{\text{anneal3}} = \frac{\langle H_i\rangle_T - \langle H_i\rangle_0 + \langle H_f\rangle_0 - \langle H_f\rangle_T}{\|[H_f, H_i]\|}, \quad (13)$$

where $\langle H_i\rangle_T = \langle\psi_T|H_i|\psi_T\rangle$, $\langle H_f\rangle_T = \langle\psi_T|H_f|\psi_T\rangle$, $\langle H_i\rangle_0 = \langle\psi_0|H_i|\psi_0\rangle$ and $\langle H_f\rangle_0 = \langle\psi_0|H_f|\psi_0\rangle$. For the example considered in the previous paragraph, $\|[H_f, H_i]\| \approx h_{\max}|V| = h_{\max}n$. Moreover, $\langle H_i\rangle_T - \langle H_i\rangle_0 = \mathcal{O}(n)$ and $\langle H_f\rangle_0 - \langle H_f\rangle_T = \mathcal{O}(n)$. Thus, the bound (13) yields $T \geq \tau_{\text{anneal3}} = \mathcal{O}(1)$. Next, we show a second example where the new bound (9) is tighter than the ones previously considered in the literature.

### B. Case study: a perturbed $p$-spin model

The $p$-spin model is described by the Hamiltonians

$$H_i = n\left(\mathbb{1} - \frac{M_x}{n}\right) \quad \text{and} \quad H_f = n\left(\mathbb{1} - \frac{M_z^p}{n^p}\right), \quad (14)$$

where $M_z = \sum_{j=1}^n \sigma_z^{(j)}$ and $M_x = \sum_{j=1}^n \sigma_x^{(j)}$ are the spins' magnetizations along $z$ and $x$, respectively [37, 38].

The free parameter $p$ determines the energy gap between the ground and first excited states which, in turn, determines the timescales to anneal adiabatically [39]: $\tau_{\text{adiabatic}} \sim \text{poly}(n)$ for $p = 2$ and $\tau_{\text{adiabatic}} \sim \exp(n)$ for $p \geq 3$ [39]. Optimized schedules significantly beat the times to anneal adiabatically; a depth-2 QAOA can prepare the target state on a timescale $T = \mathcal{O}(n^{p-1})$ [40].

Consider a perturbed $p$-spin model where an extra, low-connectivity vertex is added to the final Hamiltonian,

$$H_f' = n\left(\mathbb{1} - \frac{M_z^p + \lambda\sigma_z^{(n)}\sigma_z^{(n+1)}}{n^p}\right), \quad (15)$$

where $\lambda \in \mathbb{R}$. The extra vertex acts like a perturbation $\propto 1/n^{p-1}$ that does not significantly change the Hamiltonian's norm. We assume the initial Hamiltonian is extended to act on the appended vertex.

Choosing $W = \sigma_x^{(n+1)}$ in (8) yields the lower bound

$$T \geq \frac{n^{p-1}}{\sqrt{2}g_{\max}\lambda}, \quad (16)$$

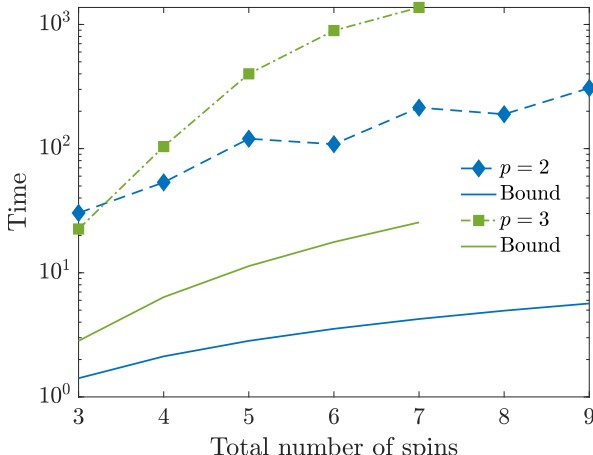

FIG. 2. Annealing times to prepare the ground state of the p-spin model with an extra node, as described by Eq. (15), as a function of the number of total spins. We consider $p = 2$ (blue diamonds) and $p = 3$ (green squares). The annealing times where obtained by numerically optimizing the fidelity to prepare the ground state using BFGS, reporting the smallest time for which the ground state is prepared with a fidelity $F > 0.99$. The corresponding bounds given in Eq. (16) are shown as blue and green lines, with $g_{\max} = 1$. The schedule-independent bound in Ref. [6] is constant for this example, polynomially looser.

for the annealing time $T$ to prepare the ground state of the modified $p$-spin Hamiltonian (15). For the $p$-psin model, the bound (13) gives a constant scaling $\tau_{\text{anneal3}} = \mathcal{O}(1)$ [6] (the perturbation to the $p$-spin Hamiltonian does not affect the bound's scaling). Eq. (16) thus shows a polynomial improvement.

In Fig. 2 we compare the bound in Eq. (16) to the times corresponding to an optimized QAOA schedule for the perturbed p-spin model [Eq. (15)]. QAOA consists of sequentially applying gates $e^{-i\beta_j H_i}$ and $e^{-i\gamma_j H_f}$ for $j = \{1, \cdots, L\}$, where the angles $\beta_j$ and $\gamma_j$ are optimized to approach the target ground state of $H_f$ [11]. We refer to Appendix C for further details. The run-times $T = \sum_j |\beta_j| + |\gamma_j|$ to prepare the ground state were obtained by numerically maximizing the fidelity $F = \langle\psi(T)|P|\psi(T)\rangle$ using the BFGS algorithms where $P$ is the projector onto the ground state for different annealing times $T$. We additionally optimized over 100 randomly chosen initial angles and report the smallest $T$ for which the ground state is prepared with fidelity $F > 0.99$ (green squares and blue diamonds). From the results shown in Fig. 2 we see that the bound (16) captures the scaling in system size of the annealing time.

The scaling in (16) is a result of the perturbation added to the $p$-spin model. The perturbation adds a minimally-connected vertex to a well-connected graph, the scenario that we identified after (12) as the most favourable for the bound derived in this work. It is not surprising that the minimum timescale may be determined by the weakest link since, in this case, the state of the appended spin can

only evolve due to the perturbation. But, we highlight that previous bounds on annealing times did not capture such behaviour, as their scaling was determined by more global properties of the defining Hamiltonians (e.g., as in Eq. (13)).

## IV. DISCUSSION

In this work, we have taken steps to address shortcomings of previous bounds on annealing times considered in the literature. Our main result, the bound (9), can take into consideration a problem's locality structure and, for optimization problems characterized by a well-connected graph with a low-connected vertex, displays a polynomial improvement over previous results. To illustrate regimes where the bounds obtained here are tighter than previous ones, we considered a perturbed annealing model where teh bound (9)'s scaling depends on the perturbation. Along similar lines, it has been noted in Ref. [41] that, if uncorrected, perturbations can strongly influence annealing processes.

Our results thus add to a series of recent works deriving runtime bounds on computing times from the physical models used to compute. When sufficiently tight, these sort of bounds can be used to benchmark concrete algorithms; an algorithm that computes saturating a bound is optimal.

In the literature of speed limits, it has been noted that the bounds often depend on the metrics chosen [15, 42]. It would be interesting to explore if the bounds derived in this letter can be improved upon by exploring other metrics in state space or observable-based.

## ACKNOWLEDGMENTS

L.P.G.P. acknowledges support from the Beyond Moore's Law project of the Advanced Simulation and Computing Program at Los Alamos National Laboratory (LANL) managed by Triad National Security, LLC, for the National Nuclear Security Administration of the U.S. DOE under contract 89233218CNA000001, the Laboratory Directed Research and Development (LDRD) program of LANL under project number 20230049DR, and the U.S. Department of Energy, Office of Advanced Scientific Computing Research, Accelerated Research for Quantum Computing program, Fundamental Algorithmic Research for Quantum Computing (FAR-QC) project. C.A. acknowledges support from the National Science Foundation (Grant No. 2231328).

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

## Appendix A: Upper bounding the Bures distance

Here we establish the upper bound for Bures distance given in (2) in the main body of the manuscript.

We consider two quantum states $|\psi(t)\rangle = U(t)|\psi_0\rangle$ and $|\tilde{\psi}(t)\rangle = \tilde{U}(t)|\psi_0\rangle$ whose unitary evolutions $U(t)$ and $\tilde{U}(t)$ are generated by the Hamiltonians $H(t)$ and $\tilde{H}(t)$, respectively. Since [20],

$$U^\dagger(t)\tilde{U}(t) - \mathbb{1} = i\int_0^t U^\dagger(t')(H(t') - \tilde{H}(t'))\tilde{U}(t')\, dt', \tag{A1}$$

the Euclidian distance between the two quantum state is given by

$$\| \, |\tilde{\psi}(t)\rangle - |\psi(t)\rangle \, \| = \|(U^\dagger(t)\tilde{U}(t) - \mathbb{1})|\psi_0\rangle\| = \left\| i\int_0^t U^\dagger(t')(H(t') - \tilde{H}(t'))\tilde{U}(t')|\psi_0\rangle \, dt' \right\|,$$

where we used unitary unitary invariance of the Euclidian vector. As $\| \, |\tilde{\psi}(t)\rangle - |\psi(t)\rangle \, \| \geq D_B(|\psi(t)\rangle, |\tilde{\psi}(t)\rangle)$ is lower bounded by Bures distance $D_B$, upper bounding the right-hand side and using unitary invariance again establishes the desired result.

$$D_B(|\psi(t)\rangle, |\tilde{\psi}(t)\rangle) \leq \left\| i\int_0^t U^\dagger(t')(H(t') - \tilde{H}(t'))\tilde{U}(t')|\psi_0\rangle \, dt' \right\| \leq \int_0^t dt' \left\| (H(t') - \tilde{H}(t'))\tilde{U}(t')|\psi_0\rangle \right\| \tag{A2}$$

## Appendix B: Deriving the bound (8)

To derive the bound (8) in the main text, we further upper the right-hand side of (A2) to obtain

$$D_B(|\psi(t)\rangle, |\tilde{\psi}(t)\rangle) \leq \int_0^t \|(H(t') - \tilde{H}(t'))\| \, dt', \tag{B1}$$

where here $\| \cdot \|$ denotes the spectral norm. Choosing $\tilde{H}(t) = W^\dagger H(t)W$ where the unitary $W$ satisfies $[W, H_i] = 0$ and $W|\psi_0\rangle = e^{i\phi}|0\rangle$ gives

$$|\langle\tilde{\psi}(t)|\psi(t)\rangle| = |\langle\psi_0|W^\dagger U^\dagger(t)WU(t)|\psi_0\rangle| = |\langle\psi(t)|W|\psi(t)\rangle| \tag{B2}$$

the desired result.

## Appendix C: Bounding the number of layers in QAOA

QAOA consists of sequentially applying gates $e^{-i\beta_j H_i}$ and $e^{-i\gamma_j H_f}$ for $j = \{1, \cdots, L\}$ [11]. The angles $\beta_j$ and $\gamma_j$ are usually optimized through a classical optimization routine, in tandem with a quantum computer to minimize the expectation value of the Hamiltonian $H_f$. The goal is to prepare a state close to the ground state of $H_f$ with as few layers $L$ as possible.

QAOA can be interpreted as an annealing problem where the annealing schedule is of the form $f(t) = 1 - g(t)$, and $g(t)$ switches between the values $g(t) \in \{0, 1\}$ to alternate between applying the unitaries $e^{-i\beta_j H_i}$ and $e^{-i\gamma_j H_f}$. In this setting, the angles in QAOA correspond to the time intervals each Hamiltonian acts for in the quantum annealing algorithm.

Following the steps taken in Ref. [7], the lower bounds (8) and (9) on annealing times imply lower bounds on number of layers $L$ needed for QAOA to prepare the desired ground state. We first note that a bound on annealing times implies a bound on the sum of QAOA angles. From (9) we obtain

$$\sum_{j=1}^L (|\beta_j| + |\gamma_j|) \geq \max\left\{ \frac{D_B(|\psi_0\rangle, V|\psi_0\rangle)}{\|[H_i, V]\|}, \frac{D_B(|\psi_T\rangle, W|\psi_T\rangle)}{\|[H_f, W]\|} \right\}. \tag{C1}$$

For periodic Hamiltonians with $\{|\beta_j|, |\gamma_j|\} \in [0, 2\pi]$ [7], the previous expression yields a lower bound on the number of layers

$$L \geq \max\left\{ \frac{D_B(|\psi_0\rangle, V|\psi_0\rangle)}{4\pi\|[H_i, V]\|}, \frac{D_B(|\psi_T\rangle, W|\psi_T\rangle)}{4\pi\|[H_f, W]\|} \right\} \tag{C2}$$

needed to prepare the ground state of $H_f$.

The latter expression lower bounds the circuit depth of any QAOA. It can be used to benchmark the optimization procedure; i.e., there is no more need to continue optimizing the angles $\beta_j$ and $\gamma_j$ for QAOA protocols where the system's state is sufficiently close to the target state.