# Peer review of "Tighter Lower Bounds on Quantum Annealing Times"

_SciPost Physics_

## Round 1 · Referee Report · Anonymous (Referee 1) · 2025-1-5

Strengths

The manuscript improves a lower bound on the anealing time necessary to prepare the ground state of a target Hamiltonian. The bound is easy to evaluate because it does not depend on the annealing schedule.

Weaknesses

A natural question is how tight is the bound. The question is answered for a few examples where it is shown to be quite tight. Given the circumstantial nature of the evidence, I am wondering how general is this conclusion. In particular, does it apply to the classic example of the Kibble-Zurek ramp in the transverse field quantum Ising chain, see e.g. the review https://arxiv.org/pdf/0912.4034? The chain is ramped from a purely transverse field to a purely ferromagnetic Ising chain across a quantum phase transition. For a linear ramp the minimal annealing time T \propto N^2, where N is the number of spins. It can be reduced to T \propto N^1 for a non-linear ramp, see section 2.5. In contrast, Eq. (5) seems to predicts T \geq 1/N that is correct but not tight.

Report

I would like the authors to include my counter-example in the manuscript and elaborate on the assumptions that have to be satisfied in order to make their equation (5) a tight lower bound.

Recommendation

Ask for major revision

  • validity: good
  • significance: low
  • originality: ok
  • clarity: good
  • formatting: good
  • grammar: excellent

Author:  Luis Pedro Garcia-Pintos  on 2025-03-10  [id 5274]

(in reply to Report 1 on 2025-01-05)

We thank the Referee for the time dedicated to reviewing our submission, and we are glad that they found it to be timely and clearly written.

We also thank the Referee for highlighting the importance of the tightness of our new bounds. Indeed, deriving bounds tighter than those previously known was the main motivation behind our work. Moreover, we agree with the Referee's conclusion that Eq. (5) is very loose for the transverse field Ising chain. We have added an entirely new section, Sec. IV, where we clarify this. In the new section, we also clarify that the other main result of our manuscript, Eq.~(9) in Sec. III, provides bounds generally tighter than Eq.~(5) for spin chains. For the example mentioned by the Referee, Eq.~(9) yields a constant bound, polynomially better than Eq.~(5). Eq.~(9) is still loose for the transverse field Ising chain, but becomes tighter for sufficiently connected graphs [see Eq. (12) and the discussion that follows].

The Referee will find a version of the new manuscript where these new additions, as well as other changes in response to the other referee reports, are marked in color.

Attachment:

main_marked_changes-compressed_lRB84AG.pdf

---

## Round 1 · Referee Report · Anonymous (Referee 2) · 2025-2-2

Report

This is a report on the manuscript entitled ``Tighter Lower Bounds on Quantum Annealing Times'', by L. P. Garcia-Pintos et al.

The manuscript main purpose is to provide lower bounds on the minimal time needed for a Quantum Annealing algorithm to drive a simple initial state, typically the state $|\psi_0\rangle$ with all spins aligned in the x-direction, into the target ground state of some problem Hamiltonian $\hat{H}_f$.

The main results of the paper are, if I understand correctly:

1) Eq. (3) and its consequence Eq.~(5), which provides a schedule-independent lower bound:

$$ T \ge \max \left{ \frac{D_B(|\psi_T\rangle,|\psi_0\rangle)}{g_{\max}\sqrt{\mathrm{Var}{|\psi_0\rangle}(H_f)}},
\frac{D_B(|\psi_T\rangle,|\psi_0\rangle)}{f
\right} \hspace{20mm} \mathrm{Eq.}(5) $$}\sqrt{\mathrm{Var}_{|\psi_T\rangle}(H_i)}

2) Eq. (9), which, for spin networks on graphs, allows to deduce bounds of annealing times scaling with the network connectivity, at variance with results present in the literature, Ref. 6, completely insensitive to the connectivity. \end{description}

I find that the physics described in the paper is sound, the math quite well explained, and the presentation well organized. Nevertheless, I do have concerns about how tight such improved ``tighter lower bounds'' in the end are.

Particularly noteworthy is the size dependence of the bound in Eq.~(5), which is derived from Eq.~(3) with the usual assumption that the time-dependent Hamiltonian $H(t)=f(t) H_i + g(t) H_f$, where $H_i$ is the Hamiltonian with the ``simple to construct ground state'' $|\psi_0\rangle$, and $H_f$ is the problem Hamiltonian of which we want to construct the ground state. If you apply such bound to a general two-spin Hamiltonian,

$$ H_f = -\sum_{\langle i,j\rangle} J_{ij} \hat{\sigma}^z_i \hat{\sigma}^z_j \;, $$
with $H_i=-\sum_j \hat{\sigma}^x_j$ (the standard transverse field term), one gets (if I have done no mistake in my calculation):
$$ \sqrt{\mathrm{Var}{|\psi_0\rangle}(H_f)} = \sqrt{ \sum\;. $$} J^2_{ij}

For any short-range (e.g., nearest-neighbor couplings of any sign on any lattice), this quantity scales as $\sqrt{N}$, where $N$ is the number of spins in the lattice. As a consequence, the bound in Eq.(5) tells us that
$$ T \ge \frac{D_B(|\psi_T\rangle,|\psi_0\rangle)}{g_{\max}\sqrt{\mathrm{Var}_{|\psi_0\rangle}(H_f)}} \propto \frac{1}{\sqrt{N}} \;. $$
So, the bound on the time decreases for increasing $N$. This is clearly very unsatisfactory as a tighter bound: even a uniform Ising chain is known to require a time $T$ scaling as $N$, not to mention Ising problems with frustration, where the time $T$ is presumably scaling much worse.

So, my question is: what should we do with Eq.~(5) in problems that are not Grover-like (infinite range) where the correct scaling with $\sqrt{d}$, Eq.~(7), follows, because the variance decreases as $1/\sqrt{d}$?

I think that the authors should address this point in some way.

Other minor things.

  1. No reference is given concerning the Bures distance in Eq.(1). I believe that some references should be added.

  2. On page 4, left column, there is a spelling problem: teh bound'' $\to$the bound''.

  3. In appendix B, right above Eq.(B2), I think that in $W|\psi_0\rangle=e^{i\phi}|0\rangle$, the $|0\rangle$ should be substituted by $|\psi_0\rangle$.

  4. Eq. (C2) is derived for periodic Hamiltonians, where the QAOA angles can be taken in $[0,2\pi]$. Nevertheless, the sentence immediately below (``The latter expression lower bounds the circuit depth of any QAOA.''), suggests a more general application. I would ask the authors to rephrase that sentence appropriately.

Recommendation

Ask for major revision

  • validity: -
  • significance: -
  • originality: -
  • clarity: -
  • formatting: -
  • grammar: -

Author:  Luis Pedro Garcia-Pintos  on 2025-03-10  [id 5273]

(in reply to Report 2 on 2025-02-02)

We are grateful to the Referee for the attention they dedicated to our manuscript, as well as for the feedback. We are happy that the Referee judges that ``the physics described in the paper is sound, the math quite well explained, and the presentation well organized''. We have responded to all of the referee's suggestions below.

We thank the Referee for highlighting the importance of the bounds' tightness. The concern with the tightness of previous bounds in the literature was our main motivation behind the current work. We agree with the Referee's conclusion that Bound (5) is loose for spin chains. Given this, we added Section IV to discuss the looseness of the bound for spin chain Hamiltonians. In the new section, we also highlight that Eq. (9) is significantly tighter, yielding a bound constant in system size. Please see the paragraph before Eq. (13), which corresponds to the same Hamiltonian model suggested by the Referee. Note that, more generally, Eq. (12) provides bounds that depend on the connectivity of the graph that describes the Hamiltonian.

Regarding the other details highlighted by the Referee:

(1) We have added citations to the PhD thesis by Chris Fuchs and to the book by Nielsen and Chuang.

(2-4) Thank you for spotting these typos, we have fixed them all, along with other minor typos we found.

We thank the referee for the exhaustive work and the constructive feedback. All changes made in the manuscript are highlighted in blue.

Attachment:

main_marked_changes-compressed_9w9nHKR.pdf

---

## Round 1 · Referee Report · Anonymous (Referee 3) · 2025-2-10

Report

The manuscript by Garcia-Pintos and colleagues introduces new quantum speed-limit times for quantum annealing. The main result, Eq. (9), does not depend on the annealing schedule (and is thus easy to evaluate), can take the local structure of the Hamiltonian into account, and is polynomially tighter than existing bounds. The paper builds up on the recent PRL (Ref. [6]) by one of the authors, together with the techniques of Refs. [20,21] by another author. The theory is applied to spin networks and to a perturbed p-spin model (see Fig. 2).

I enjoyed reading the manuscript which is timely and clearly written. I have three questions:

1) the authors correctly mention at the end of the paper that speed limits usually depend on the choice of the metric. However, they do not provide any justification for the use of the Bures distance in the present study.

2) One of the main observations of Ref. [6] is the existence of a trade-off between speed limit and coherence. However, the role of quantum coherence is not mentioned at all in the current analysis. Does it help to make the bounds tighter (or does it make them looser)?

3) The bound of Ref. [6] is constant for the example of Fig. 2. I think, it would be helpful to explicitly show it in the figure (the reader has no idea of the value of that constant, and thus cannot really judge how much much tighter the new bounds really are).

Recommendation

Ask for minor revision

  • validity: high
  • significance: high
  • originality: high
  • clarity: high
  • formatting: excellent
  • grammar: excellent

Author:  Luis Pedro Garcia-Pintos  on 2025-03-10  [id 5272]

(in reply to Report 3 on 2025-02-10)

We thank the referee for the positive outlook on our paper. Regarding the three questions:

(1) We added a sentence after Eq. (1) that refers to the operation meaning of the Bures distance, which partly explain why it is often used to study speed limits. Specifically, the new draft includes:
"While one could consider other metrics, we adopt the Bures distance given its operational meaning: the Bures distance upper and lower bounds the likelihood to distinguish between two states given any possible measurement on the system [REF Nielsen Chuang]."

(2) Thank you for the very interesting question. While it is likely that tighter bounds than the ones we have derived in the current submission could be obtained (e.g., by accounting for the system's coherence), the bounds in Ref. [6] require information about the system's evolution to be evaluated. In particular, evaluating the system's coherence throughout the computation requires knowledge of its state at all times. Thus, bounds that involve the coherence would go against the aim of the current submission, where we specifically focus on bounds that can be evaluated without information of the algorithm's schedule or the state of the system. We added the following sentence before Eq. (13):
"(Note that the tightest bound derived in Ref.~[6], which involves the computer's coherence along the computation, depends on the time-evolved state of the system.)"

(3) Thank you for the suggestion. We added the schedule-independent bound from Ref.~[6] to figure 2, confirming that the new bounds are significantly tighter. We also added the following sentence to the figure's caption:
"The schedule-independent bound in Ref.~[6] is shown for this example as a dashed line."

Once again, we thank the referee for the suggestions. All changes to the manuscript are marked in color in an appended document.

Attachment:

main_marked_changes-compressed.pdf

---

## Editorial Decision

resubmitted